Decoding the bare necessities of decapod crustacean nomenclature through the ages

http://orcid.org/0000-0002-2437-2445 De Grave Sammy 1 sammy.degrave@oum.ox.ac.uk
Cole Elizabeth 1
http://orcid.org/0000-0002-3759-8945 van der Meij Sancia E. T. 2 3
1 Oxford University Museum of Natural History , Oxford , United Kingdom
2 Naturalis Biodiversity Center , Leiden , Netherlands
3 GELIFES, University of Groningen , Groningen , Netherlands
Badenhorst Shaw
Electronic publication date: 2025 Nov 11
Publication date: 2025
Volume: 13
Electronic Location ID: e20337
Received 2025 Aug 21; Accepted 2025 Oct 13
Copyright: © 2025 De Grave et al.
Copyright year: 2025
Copyright holder: De Grave et al.
License: This is an open access article distributed under the terms of the Creative Commons Attribution License, which permits unrestricted use, distribution, reproduction and adaptation in any medium and for any purpose provided that it is properly attributed. For attribution, the original author(s), title, publication source (PeerJ) and either DOI or URL of the article must be cited.
License URL: https://creativecommons.org/licenses/by/4.0/

Keywords: Biodiversity, Decapoda, Etymology, Taxonomy, Systematics

Funding: Research Development Grant from the Digital Humanities Institute at the University of Oxford This research was funded by a Research Development Grant from the Digital Humanities Institute at the University of Oxford. The funders had no role in study design, data collection and analysis, decision to publish, or preparation of the manuscript.

==============================
Though taxonomists have been classifying species since 1758, the methods and biases of their naming practices have recently come under scrutiny. Despite some compelling claims on e.g., historical imperialism in the published literature, the knowledge base for making such assertions is small, as nomenclatural trends have only been researched in a select few taxa. Here, we investigate naming practices in Decapoda, one of the most studied crustacean groups, thereby extending the knowledge base to the marine realm in contrast to a previously studied cohort of largely terrestrial taxa. To date almost 18,000 species of decapods are known, from which a total of 22,363 unique names are analysed, as neither nomenclatorial nor taxonomic status has any bearing on the naming process. Despite taxonomists being inspired by a multitude of cultural influences, historically the majority of names were derived from the morphology of the animals. This dominance declined in the Victorian era, with a concomitant rise in the use of both geographically inspired names and eponyms (species named after people). Post-1958, a near-even split is achieved between these three categories, while other etymological classifications stake a minority claim on the dataset. Although a historic and contemporary gender imbalance is present amongst eponyms honouring scientists, contrary to previous findings our results detect no actual bias in naming practices, instead indicating that female scientists have been honoured in proportion to their collective presence in the field. Though previous studies have flagged a significant proportion of eponyms named for colonialist figures, these were found in relatively small numbers among Decapoda.

Introduction

Taxonomists have been describing and naming species for over 250 years, with the official starting point of zoological nomenclature being the 12th edition of Linnaeus’ ‘Systema Naturae’ (Linnaeus, 1758). Although names must adhere to the rules of the International Code of Zoological Nomenclature (ICZN, 1999), these rules are largely concerned with the formation of the names, rather than their derivation or meaning, which are largely free of constraints. Throughout the decades taxonomists have used a multitude of sources for inspiration (see e.g., Jóźwiak, Rewicz & Pabis, 2015), ranging from distinctive morphologies of the taxon in question through to Greek mythology, Norse legends, classic literature, musical icons and so on. Many species also have eponymous names, after famous scientists, often working on the same general group, people involved in the collecting or descriptive process or family members. Names of species should not be considered trivial; for example, Mlyranek et al. (2023) show that phytophagous arthropods feature more heavily in certain lines of research if named after their host plant. Equally, Blake et al. (2023) drew attention to the fact that species from a selection of taxa (invertebrates, amphibians, reptiles, etc.,) that were named for present day celebrities, e.g., Aphonopelma johnnycashi Hamilton, Hendrixson & Bond, 2016, received many more Wikipedia page views, and thus more exposure to the general public.

It remains rather unclear whether the inspiration for new names is largely influenced by intrinsic features of the taxa in question (as may be expected when using morphology-based names), the people studying them (e.g., Roger Bamber who named numerous species of tanaids for characters in Terry Pratchett’s Discworld series; see Jóźwiak, Rewicz & Pabis, 2015) or temporal trends and fashions. In recent years, a number of studies have appeared which address this knowledge gap. Poulin, McDougall & Presswell (2022) investigated naming trends for approx. 2,900 parasitic helminths, described between 2000 and 2022. Their study revealed considerable bias depending on the higher taxonomic group, with for example nematodes named more often after an eminent scientist than a morphological feature, the reverse being the case in acanthocephalans. Equally, an increasing trend was identified in naming species after family or friends over the period 2000–2022, with the suggestion proffered this should be avoided in future, as the authors consider it too close to self-naming, a practice generally frowned upon by taxonomists. The most comprehensive study to date in this respect is by Mammola et al. (2023) using in excess of 48,000 names of spiders, in a dataset spanning from 1757 to 2020. A clear trend was identified with morphologically-inspired etymologies peaking in 1850–1900, followed by a steady decline and a parallel increase in etymologies dedicated to people or geography. Their study also identified an increase in pop culture references in the period 2000–2020, blamed on the need for taxonomy (and taxonomists) to remain visible in today’s scientific climate. Since then, only two further studies have appeared, each on relatively small datasets. Pardos & Cepeda (2024) analysed the marine, meiobenthic Kinorhyncha with 421 species, spanning 1863–2024. The observed trends were in general agreement with previous studies, although it was emphasised that each author follows their own nomenclatorial path in terms of style and preferences. Kazanidis (2024) looked at 425 genera of Echinodermata and concluded that after 1960 the usage of morphologically-derived etymologies declines, with a parallel increase in using scientists’ surnames.

Although the merits of comprehensive exercise in unravelling such etymological trends may be more subtle than those of flashier studies, that’s not to say they are not there. Such work draws attention to the nuances and eccentricities of the rather unglamourised but vital field of taxonomy, and explores topical aspects of academic culture such as gender imbalance and historic imperialism. Within etymological analysis, it is clear that the knowledge base is not yet broad enough to draw overarching conclusions. Headline chasing studies, like Guedes et al. (2023), who argued that many historical eponyms are problematic, can therefore not be fully put in perspective, and their conclusions may well be exceptions rather than the norm. To add to the knowledge base, we herein analyse a comprehensive data set (17,719 recent species and 22,363 unique names) of Decapoda, taxonomically well-studied through the ages, but also of considerable conservation, cultural and dietary importance.

Materials and Methods

The starting point of the analysis was a download from DecaNet (De Grave et al., 2023) of all available and unavailable (sensu ICZN) species-level (species, subspecies) and infrasubspecific names (variety, forma, natio) published from 1758 up to end 2024; fossil taxa were excluded from the dataset. As neither nomenclatorial nor current taxonomic status has any bearing on how names are constructed, the dataset not only comprises currently accepted species-level names, but also junior subjective synonyms, junior homonyms (e.g., Cancer longipes Bell, 1835 a junior homonym of C. longipes Linnaeus, 1758), nomina dubia (e.g., Palaemonetes natalensis Stebbing, 1915), nomina nuda (e.g., Coenobita compta White, 1847), as well as unavailable names (e.g., Potamon (Centropotamon) hueceste hueceste natio agris Pretzmann, 1983). For a definition of these categories, see the International Code of Zoological Nomenclature (ICZN, 1999) and Horton et al. (2017). For all entries the original spelling was coded for, irrespective of grammatical agreement and whether the spelling is mandatory (sensu ICZN) to correct, e.g., Cancer (Xantho) 5-dentatus Krauss, 1843. The total dataset comprises 22,363 entries.

Names were classified into seven broad categories (some with subcategories), following the scheme outlined in Mammola et al. (2023), viz. ‘Morphology’, ‘Ecology’, ‘Geography’, ‘People’, ‘Culture’, and ‘Other’, to which was added ‘Expeditions’. To assign etymologies, we first checked the original descriptions, where post-1950 it is common to discuss etymology and post-1970, routinely so. For articles lacking an etymology section, the whole text was scanned for clues as to the origin of the name. For those descriptions lacking any information (standard for pre-1900 descriptions), etymology was inferred based on our knowledge of Greek and Latin, with the help of standard dictionaries, internet searches and assistance from colleagues.

The category ‘Morphology’ was used when the etymology referred to the size of the species (subcategory ‘Size’, e.g., Caridella minuta Calman, 1906; Mathildella maxima Guinot & Richer de Forges, 1981), the shape of the body or some body part (subcategory ‘Shape’, e.g., Spirontocaris brachydactyla Rathbun, 1902; Medaeus latifrons Chace, 1942) or the general aesthetic/appearance of the species (subcategory ‘Colour’, e.g., Mursia flamma Galil, 1993; Caridina alba Li & Li, 2010).

As per (Mammola et al., 2023), the category ‘Ecology’ was used when the etymology referred to some aspect of the ecology or habitat of the species (subcategory ‘Habitat’, e.g., Callianassa profunda Biffar, 1973; Alpheus saxidomus Holthuis, 1980) or some behavioural aspect (subcategory ‘Behaviour’, e.g., Raninoides fossor A. Milne-Edwards & Bouvier, 1923; Cherax destructor Clark, 1936). As numerous decapods have symbiotic lifestyles, the subcategory ‘Host’ was added, used when the name clearly referred to the host species or higher systematic group, e.g., Ostracotheres spondyli Nobili, 1905; Synalpheus spongicola Banner & Banner, 1981.

Etymologies referring to the distribution of the species, irrespective of how vague (e.g., Gebiacantha arabica Ngoc-Ho, 1989) or precise (e.g., Hamopontonia essingtoni Bruce, 1986) were coded as ‘Geography’.

The category ‘People’ was used when the etymology was dedicated to a scientist or person involved in the collection or descriptive process (subcategory ‘Scientists’, e.g., Thalamita stimpsoni A. Milne-Edwards, 1861; Paratymolus apeli Naderloo & Türkay, 2015), or else other people who do not meet these criteria, most often family members (subcategory ‘Other People’, e.g., Lithodes rachelae Ahyong, 2010; Odontozona edyli Criales & Lemaitre, 2017).

Fictitious people (e.g., Periclimenes rincewindi De Grave, 2014) were coded under ‘Culture’. The category ‘Culture’ includes references to mythology, pop culture, musical bands and so on. The subcategories ‘Modern Culture’ and ‘Past Culture’ were relative to the description. For example, Garthambrus darthvaderi McLay & Tan, 2009, the eponymous Star Wars villain, was coded as ‘Modern Culture’, whilst Nephrops neptunus Bruce, 1965, the Greek god of the sea, was coded as ‘Past Culture’.

The category ‘Expeditions’, included taxa named after the expedition vessels, e.g., Hemipagurus albatrossae Asakura, 2001 or the expeditions themselves, e.g., Euryxanthops cepros Davie, 1997.

Any names which did not fit into any of the above categories were assigned to the category ‘Other’. This included arbitrary combinations of letters, anecdotes, but also derivations such as affinis, e.g., Myra affinis White, 1847, from the Latin ‘closely related to’, or typicus, e.g., Pterocaris typica Heller, 1862, meaning typical for the genus.

As scoring was carried out by all three authors, a cross-validation was carried out for a randomly selected 100 taxa. Agreement was high: 100% for categories and 89% for subcategories, with the main discrepancy being between ‘Morphology: Shape’ and ‘Morphology: Colour’. A further cross validation was carried out to check the validity of the inferred etymologies (representing an estimated 54% of the dataset). For this a randomly selected 200 taxa were selected and each scored, a posteriori, for whether our inference; (1) matched, i.e., either a stated etymology was present or the inferred was supported by in-text information; (2) did not match, i.e., there was evidence in the text for another explanation or (3) neither matched nor mismatched, i.e., nothing to invalidate the inferred etymology. Of these, 90% matched our inference, 10% neither matched nor mismatched, and none mismatched.

An analysis at subcategory level was deemed to be of limited value, especially owing to discrepancies among categorisation at this level, with the exception of a further gender analysis of the subcategory ‘People: Scientists’ (see below). Given the cross-validation results, we assume the list to only contain trivial errors and proceed with the analysis at category level.

Annual sum count and proportion were calculated for each etymology category. A generalised additive model was applied to temporal trends in the proportion of each etymology category, assuming a quasibinomial distribution and a logit link function. All data processing was carried out in R v4.5.0 (R Core Team, 2025), using ggplot2 v3.5.2 (Wickham, 2016) for visualisation.

In contrast to Mammola et al. (2023), only single meanings were allowed. All information has been uploaded to the DecaNet portal of WoRMS (www.decanet.info) under the tab ‘Attributes’, sub-tab ‘Etymology’. Throughout the text we largely cite the taxa in their original orthography and generic combination, supplemented, if necessary, by their current status and generic affiliation. To discuss temporal trends, we refer to the framework of descriptive taxonomic effort established by De Grave et al. (2023), which identified five distinct periods in the accumulated knowledge of decapod taxonomy, viz. the ‘Wunderkammer’ era (1759–1836), ‘Victorian’ era (1838–1913), ‘World in turmoil’ era (1914–1955), ‘Sputnik’ era (1958–2000) and the ‘New taxonomy’ era (2002–present day).

In order to investigate potential gender bias in naming practices, a further analysis of the ‘People: Scientists’ subcategory was carried out. According to the ICZN nomenclatorial rules, when naming a species after a person (eponym), the ending must reflect the gender of the honouree. If female, then the ending should be -ae, for example Synalpheus dorae Bruce, 1988 named after Dora Banner; if male then the ending is usually -i, for example Goneplax clevai Guinot & Castro, 2007, named for Régis Cleva. This rule was used as a guide to deconstruct the ‘Scientists’ subcategory by gender for the subset of species from 1958 onwards, to focus on contemporary eponymic naming practices. Nomenclature is, however, far from perfect, and decapods are no exception. Many other honorific constructs exist, for example unaltered names like Parasesarma chiahsiang Shih, Hsu & Li, 2023 named for Chia-Hsiang Wang, or the grammatically incorrect Porcellana gordoni Johnson, 1970 named for Isabella Gordon. Eponyms raising reasonable doubt over the accuracy of the gendered ending or lacking it altogether were manually checked and validated, aided by stated or inferred etymologies in the descriptions as well as contextual knowledge. As far as possible, variations of names honouring the same individual were synonymised to determine eponym counts per individual scientist. On the other side of the coin, names which could refer to more than one individual (e.g., edwardsi which could refer to either Henri Milne Edwards or Alphonse Milne-Edwards) were likewise investigated and differentiated, where possible. These data were then analysed in terms of honorific naming by individual scientists.

Results and discussion

Etymologies in numbers

The total dataset consisted of 22,363 entries with 11,981 unique etymologies. This count, however, differentiates between variations of the same name to ensure gender agreement with the generic name, e.g., Travancoriana granulata Pati & Sharma, 2013 vs. Engaeus granulatus Horwitz, 1990. The true number of unique etymologies will thus be slightly lower.

Across the entire time period (1758–2024), the majority of etymologies referred to the morphology of the taxa (43.3%), whilst a significant proportion also referred to people (24.2%) and geography (18.7%) (Fig. 1). Relatively infrequently used categories are ‘Culture’ and ‘Expeditions’. The ‘Other’ category appears somewhat large but perhaps was artificially inflated by our inability to accurately assign a number of, usually older, names.

Figure 1 Total number (1758–2024) of etymologies.

Etymologies are displayed by category as both raw counts and relative percentages of the overall dataset.

Despite almost half of all etymologies referring to a morphological aspect of the species, the majority of most frequently deployed names are from different categories. The two most frequently used etymologies across the entire dataset are from the ‘Other’ category: intermedia/intermedius (used for 91 taxa) and affine/affinis (85 taxa), followed by two from the ‘Geography’ category: japonica/japonicus (81), and orientale/orientalis (70). Only the fifth most used is from ‘Morphology’: gracilis (69), followed in the top ten by crosnieri and holthuisi (‘People’, 64 each), indica/indicus (‘Geography’, 59), and the morphological terms longipes (54), inermis (53), and gracilipes (51).

At a category level, the most used etymologies for ‘People’ reads like a Who’s Who of decapod taxonomy (Table 1), with the highest number of honorifics for the late A Crosnier (1930–2021) and LB Holthuis (1921–2008), but also recognising the monumental contributions of earlier (e.g., A Alcock, 1859–1933; JG De Man, 1850–1930) as well more contemporary taxonomists (e.g., AJ Bruce, 1929–2022). Two of the highly honoured taxonomists are female, viz. MJ Rathbun (1860–1943) and D Guinot (1933–), the latter being the only living person in Table 1, testimony to her current influence on the field.

Table 1 Ten most used etymologies per category.

Note that adjectives are counted as the same etymology, as stated.

Morphology	People	Geography	Ecology	Culture	Expeditions	Other	
gracilis (69)	crosnieri (64)	japonica/japonicus (81)	profunda/profundus (19)	diomedeae (7)	sibogae (34)	intermedia/intermedius (91)	
armata/armatus (59)	holthuisi (64)	orientale/orientalis (71)	spongicola (10)	lar (6)	investigatoris (24)	affine/affinis (85)	
longipes (54)	alcocki (45)	indica/indicus (59)	corallicola (10)	neptunus (5)	talismani (12)	typicus/typica (41)	
inermis (53)	chacei (44)	africana/africanus/africanum (49)	fluviatile/fluviatilis (10)	miles (4)	albatrossae (10)	similis (30)	
gracilipes (51)	edwardsi/edwardsii (42)	pacifica/pacificus (47)	cavernicola/cavernicolus (9)	acherontis (3)	karubar (8)	dubius/dubia (26)	
brevirostris (49)	rathbunae/rathbuni (42)	australe/australis (46)	pelagica/pelagicus (9)	arethusa (3)	challengeri (7)	dispar (18)	
laevis (48)	foresti (30)	sinense/sinensis (42)	insulare/insularis (9)	aries (3)	panglao (7)	consobrinus (15)	
elegans (46)	brucei (28)	occidentalis (40)	fossor (8)	hebes (3)	valdiviae (7)	vicina/vicinus (12)	
longirostris (44)	demani (28)	australiense/australiensis (36)	commensalis (7)	eulimene (3)	zacae (5)	debilis (10)	
spinosus (38)	guinotae (26)	atlantica/atlanticus (32)	pugnax (7)	triton (3)	hassleri (5)	variabilis (10)	

The most frequently deployed etymology in the ‘Geography’ category belongs to taxa with their type locality in Japanese waters (Table 1), whilst taxa discovered in China also feature at position seven. The others are all very broad geographical terms, e.g., Brachynotus atlanticus Forest, 1957 and Palaemonella orientalis Dana, 1852.

In terms of Ecology (Table 1) it is not surprising that terms reflecting the deeper-water, or pelagic habitat feature heavily, e.g., Stylodactylus profundus Cleva, 1990; Nectoceras pelagica Rafinesque, 1817. Given the propensity of many decapods to form symbiotic relationships with a variety of taxa, sponge- (e.g., Periclimenaeus spongicola Holthuis, 1952) and coral- (e.g., Galathea corallicola Haswell, 1882) dwelling taxa were frequently named for this relationship, whilst further derivations such as Nematopagurus spongioparticeps McLaughlin, 2004 and Galathea coralliophilus Baba & Oh, 1990 abound.

Although overall, not that frequently deployed, in ‘Culture’ all of the most frequently deployed etymologies derive from Greek mythology, many, but not all, connected to the marine realm (e.g., Alpheus neptunus Dana, 1852; Paguristes triton McLaughlin, 2008). A disproportionate number of cultural names can be attributed to individual taxonomists: for example, E Macpherson and co-authors described 27% of all cultural names, reflective of his apparent infatuation with Greek (and other) mythology. In this way, clear signals can be picked out of an individual carcinologist’s work and impact. Alongside honorific eponyms, trademark naming practices are another way taxonomists may leave their mark and legacy within their field (see also Pardos & Cepeda, 2024; Jasper, Froehlich & Carbayo, 2015).

Unsurprisingly, in the category ‘Expeditions’, a significant proportion of taxa was named for the Dutch Siboga expedition (1899–1900), largely in a series of articles in 1905–1938 by De Man, Tesch etc., which worked up the expedition’s results, e.g., Homolomannia sibogae Ihle, 1912. The collections made by the Investigator in Indian waters (e.g., Paralomis investigatoris Alcock & Anderson, 1899), as well as the Atlantic and Mediterranean collections by the Talisman are honoured in several names (e.g., Gennadas talismani Bouvier, 1906). More contemporary expeditions organised by MNHN (Paris) are also often honoured, notably the 1991 KARUBAR cruise to Indonesia (e.g., Chaceon karubar Manning, 1993) and the 2004 PANGLAO survey in the Philippines (e.g., Stereomastis panglao Ahyong & Chan, 2008).

Although there is some evidence in other groups that the usage of non-classical language (e.g., Latin, Greek) has risen through time for a variety of taxa (Heard & Mlyrnarek, 2023; Pardos & Cepeda, 2024), these continue to be a numerical minority in decapods. To give but two examples: Notosceles pepeke Dawson & Yaldwyn, 2000, a Māori derived name of a frog crab, and Hexaplax saudade Rahayu & Ng, 2014, a Portuguese word meaning a melancholic longing for something that is absent, lost or unattainable.

Homage to homer

Whilst much has been written about the humorous and inappropriate etymologies of taxa (Lalchhandama, 2014 provides a more than thorough review) across a variety of groups, in reality the vast majority of species names across all taxa are rather simple or innocuous. This is equally the case in Decapoda, with names often referring to straightforward morphological features (e.g., Sicyonia robusta Crosnier, 2003) or colour (e.g., Mithraculus ruber Stimpson, 1871) of the species, or its type locality (e.g., Metapenaeus palaestinensis Steinitz, 1932; Raymunida iranica Osawa & Safaie, 2014); see also Table 1 for further examples.

Nevertheless, a good number of whimsical, fantastical and curious etymologies exist across Decapoda, in common with all other groups studied so far.

The two shortest names, two letters as allowed by the International Code of Zoological Nomenclature (ICZN, 1999) are Zuzalpheus ul Ríos & Duffy, 2007 (now placed in Synalpheus), a Mayan word meaning ‘inside’, referring to the sponge-dwelling habitat of the species and Potamon ou Yeo & Ng, 1998 (now placed in Indochinamon), named after its type locality ‘Nam Ou’. Conversely, the longest specific epithets belong to Synalpheus hastilicrassus var. acanthitelsoniformis De Man, 1920 (junior subjective synonym of Synalpheus hastilicrassus Coutière, 1905) and Caridina pseudogracilirostris Thomas, Pillai & Pillai, 1976 (junior subjective synonym of Caridina gracilirostris De Man, 1892), each at 20 characters long.

Whimsical examples abound amongst decapod etymologies. These include Lithoscaptus doughnut Wong et al., 2023, named for the resemblance of the host’s corallites to the sugary snack, Paragiopagurus schnauzer Lemaitre, 2006, named after Patsy McLaughlin’s favourite dog breed, and Vulcanocalliax beervana Schnabel & Peart, 2024, named after a New Zealand beer festival. Tongue-in-cheek name constructs are of course not restricted to modern times, for example Lophopanopeus somaterianus Rathbun, 1930, was based on two dactyli taken from the stomach of Eider ducks (genus Somateriana Leach, 1819).

Fantastical creatures and persona are not neglected in decapod etymologies, with Harryplax severus Mendoza & Ng, 2017 (a dual reference to two main characters in JK Rowling’s ‘Harry Potter’ series) and Odontonia bagginsi De Gier & Fransen, 2018 (alluding to the Baggins family of ‘The Hobbit’ by JRR Tolkien) as examples. Other examples of popular culture (although a minority of names at 0.12% overall) are represented by, for example, Albunea groeningi Boyko, 2002, for Matt Groening, creator of the acclaimed cartoon series ‘The Simpsons’ and ‘Futurama’. Almost no decapod taxa are named for contemporary celebrities: Thor dicaprio Anker & Baeza, 2021 and Elephantis jaggeri Klotz & De Grave, 2015 are perhaps the only examples so far, despite the positive online impact such names can potentially generate (Blake et al., 2023).

Early explorers are of course honoured in a series of names, such as Synalpheus bougainvillei Coutière, 1905, as are contemporary biodiversity hunters (e.g., Pseudocoutierea wirtzi d’Udekem d’Acoz, 2001; Petrolisthes paulayi Hiller & Werding, 2016). Nobility is equally represented, e.g., Parapilumnus leopoldi Gordon, 1934 and Macrobrachium sirindhorn Naiyanetr, 2001, named for the Belgian King Leopold III and the Thai Princess Sirindhorn, respectively, in recognition of their interest in natural history. Albert I, Prince of Monaco (1889–1922) devoted much of his life and fortune to maritime pursuits, often accompanied by biologists on his yachts. In recognition, seven taxa were named in his honour, e.g., Lithodes grimaldii A. Milne-Edwards & Bouvier, 1894. A number of taxa have also been named after the Japanese Emperor Hirohito (1901–1989), e.g., Osachila imperialis Sakai, 1963; usually based on specimens collected by the Emperor himself, a renowned hydrozoan taxonomist.

Aside from the decapod taxonomists most frequently honoured with plentiful dedications (the ‘Giants on whose shoulders we stand’; Table 1), numerous others are of course also recognised, testimony to the respect they command in the community. To list but a few contemporary examples: Troglocarcinus monodi Fize & Serène, 1956; Harrovia ngi Chen & Xu, 1992; Hymenopenaeus tuerkayi Crosnier, 1995; Dactylonia franseni Bruce, 2003; and Tomopaguropsis rahayuae Jung, Lemaitre & Kim, 2017. Equally, many of those who went before have also been honoured, e.g., Herbstia ortmanni Balss, 1924; Lissoporcellana miyakei Haig, 1981 and Nikoides danae Paulson, 1875.

Guedes et al. (2023) identified a significant proportion of African vertebrates named for individuals from the European colonial period. Though we did not discriminate by geographic origin of the species, across all decapod etymologies such constructs are apparently quite rare. Amongst the examples we were able to identify are Potamon rafflesi Roux, 1936 named for British colonial official Sir Stamford Raffles, Platyxanthus balboai Garth, 1940 after the conquistador Vasco Núñez de Balboa, and more recently, Lysmata napoleoni De Grave & Anker, 2018, for the French Emperor exiled to St. Helena.

Somewhat peculiar are etymologies in which the name refers to colloquial or indigenous names for the taxon in question. For example, Linnaeus (1758) named Cancer crangon, the European brown shrimp, simply after an ancient Greek word for shrimp. Other examples are Cancer saratan Forskål, 1775, from the Arabic word meaning crab and more recently Cambarellus moi Pedraza-Lara, Ortiz-Herrera & Jones, 2021, meaning crayfish in a local Mexican tribal language. Such pseudo-tautonyms are a way of bringing forth the language and culture of native peoples who have long known these species, and thus avoiding a complete westernisation of biodiversity records: the case for such a practice was made by Gillman & Wright (2020).

Temporal trends

A significant interaction exists between the year of description and etymology categories, as defined by proportional usage through time (Fig. 2). During the ‘Wunderkammer’ era (1789–1836), morphological etymologies dominate throughout, although declining somewhat in the later years, when etymologies based on people play a significant role. During the ‘Victorian’ era a steep decline in morphology-based etymologies can be observed, concomitant with a rise in etymologies based on people and geography; this trend continues into the ‘World in turmoil’ era, although less pronounced. During the ‘Sputnik’ era and into the ‘New taxonomy’ era matters stabilise significantly and morphology-, people- and geography-based etymologies each account for roughly 25–30% of all names. All other coded categories do not show any significant temporal trend, presumably linked to their modest contribution. A very similar temporal trend was observed for spider names, with a general decline of morphology-inspired names post-1900 and an increase in etymologies based on people and geography since then (Mammola et al., 2023). A notable difference is the predominance of geographically based etymologies in spiders in the last 10 years (Mammola et al., 2023), whilst in Decapoda morphology still dominates; presumably a taxon-specific fashion.

Figure 2 Temporal variations in the relative proportion of etymologies.

A generalised additive model was applied, showing the predicted trend and 95% confidence interval.

Looking at individual frequency curves (Fig. 3), it is again clear morphology-based etymologies (Fig. 3A) dominated during the ‘Wunderkammer’ and ‘Victorian’ eras, suffering a sharp decline during the ‘World in turmoil’ era. However, they regained their importance post-1955 and continue to be widely deployed in current taxonomy. Geography-based etymologies show a general increase in usage throughout the eras (Fig. 3B), with some notable peaks. Yokoya (1933), in an important monograph on Japanese decapods, named 22 taxa japonica/us and nipponensis, amongst other geographical names. Other authors (e.g., Balss (1933), Monod (1933), Rathbun (1933)) that year also liberally used geographical etymologies, with a further 34 taxa, causing an overall spike in the data. Post 1980, geographically inspired names gained importance, in many years being 25–30% of all etymologies.

Figure 3 Temporal variation in the frequency of etymologies.

Frequencies of etymologies by year (1758–2024), line shows 5-year moving average: (A) Morphology; (B) Geography; (C) People; (D) Expeditions (E) Culture; (F) Ecology.

An early employer of eponyms was H Milne Edwards, who honoured numerous scientists in a series of works around the 1850s, notably H. Milne Edwards (1853) in which 39 such names appear (Fig. 3C). The period 1902–1905 also sees a relatively large number of taxa honouring other scientists, through the work of De Man, Rathbun, Nobili, Alcock, Coutière and contemporaries. From the 1980s onwards honouring fellow scientists, including collectors of the type material, became relatively commonplace and in many years amounts to 25–30% of all names. If the current trend of festschrifts continues, such honorifics can be forecasted to maintain numbers, especially in localised spikes. Although there are several examples of taxa named after ‘Other People’ (i.e., non-scientists and/or uninvolved in the collecting process) early on, e.g., Ilia mariannae Herklots, 1852, such name constructs only really become popularised since the 1950s, and continued to gain popularity up to the present day, although in any given year they only comprise a minor proportion (1.6% across the entire dataset).

Etymologies using references to ‘Expeditions’, ‘Ecology’ and ‘Culture’ have always been a minor component of the naming process (Figs. 1, 2). These categories show a general upward trend in frequency through the decades (Figs. 3D–3F), but proportionately have remained fairly static (Fig. 2), suggesting the increase in occurrence is more linked to the general increase in taxon descriptions (see De Grave et al., 2023) rather than an underlying temporal trend in naming practices.

Gender imbalance

In carrying out a gender analysis into eponymic naming practices of scientists, it is necessary to disentangle an expected imbalance (product and reflection of the gender divide in the wider field due to societal factors beyond the scope of this study) from any potential active bias in the naming practices (such as male scientists amassing more eponyms per individual than their female counterparts). The number of female scientists honoured throughout the entire 1758–2024 period was 263 (11.7%) compared to 1986 males. A similar disparity has been observed in helminths (18.6% after females, see Poulin, McDougall & Presswell, 2022) and molluscs (10.6%, see Vendetti, 2022). Although this gender disparity is stark, this inequity can largely be explained by historically few women studying Decapoda at the level of professional researcher, museum curator or professor, a result of barriers which only started to be removed in the late 20th Century. When the dataset was reduced to 1958 onwards to focus on contemporary trends, the gender imbalance persisted but at a slightly diluted rate: 229 females (16.9%) compared to 1,123 males. This shift corroborates a positive inclusivity trend observed by Sangster (2025) among bird eponyms over a similar timescale, and with still a little way to go approaches known figures of the actual proportion of women within taxonomy, which vary depending on sample group between 17–28% (House of Lords, 2008; Salvador et al., 2022, respectively). The particular recognition of MJ Rathbun (see Table 1), who for temporal context was the Smithsonian’s first full-time paid female scientist, might indicate that where historically present, women’s contributions are indeed eponymised.

Regarding gender bias, an analysis of eponym counts per individual found almost no bias present in the actual naming process, with scientists, once honoured, having a near equal number of taxa named after them irrespective of gender (Fig. 4). Although a minor skew is present with only 18% of females being honoured more than once (vs. 26% of males), this largely disappears with higher numbers of honorifics, with 11% of females having been honoured more than twice (vs. 13% of males).

Figure 4 Frequency density of eponym count by individual for the category ‘People: Scientist’.

Data is split by gender for the years 1958–2024. Bins past 10 per individual not shown (2.5% of individuals).

Caveats

The temporal trends identified herein for decapod etymologies are similar to those identified for helminths (Poulin, McDougall & Presswell, 2022), spiders (Mammola et al., 2023), kinorhynchs (Pardos & Cepeda, 2024) and echinoderms (Kazanidis, 2024) and can thus be considered a true reflection of changing taxonomic naming fashions. Deducing etymologies is, however, fraught with difficulty, as unless the etymology is specified (uncommon pre-1950), an a posteriori interpretation of the author’s intention must be carried out. For example, unrecorded anecdotal epithets such as Notiax santarita Thatje, 2000, named for the wine enjoyed on the cruise during which the species was collected (S Thatje, 2025, personal communication), would remain obscure to us if the describer were uncontactable. Doubtless, many such gems are lost to history. For most etymologies related to morphology or geography, this remains straightforward, however in the other categories difficulties can be encountered. For example, Sesarma calypso De Man, 1895–1898 is herein interpreted to be named after one of the Greek nymphs, in contrast with Synalpheus calypso Ashrafi, 2024 which is stated in the etymology (Ashrafi, 2024) to have its name derived from Jacques-Yves Cousteau’s vessel.

This becomes an acute problem when investigating eponyms, as on occasion authors use an oblique reference to the person being honoured, e.g., Helice epicure Ng, Naruse & Shih, 2018, named for the late Michael Türkay, a food connoisseur as well as a brilliant decapod taxonomist. Equally, authors sometimes utilise different parts of the honouree’s name, e.g., Pagurus alaini Komai, 1998 and Alpheus alaincrosnieri Anker, 2020, both of course named for A Crosnier. As discussed already, in pursuing the gender analysis, every reasonable effort was made to disambiguate these variations, thereby revealing the true number of honorific species names per individual. As a result, the highest number of honorifics are for Lipke Holthuis (85 names), a testament to his enduring legacy and influence on decapod taxonomy. Equally, a further 12 names honour Alain Crosnier, totalling 76 taxa, evidence of his influence on decapod taxonomy. This equally applies to Danièle Guinot, who in addition to the 26 ‘guinotae’ (Table 1) is honoured in a further 14 names, e.g., Dicranodromia danielae Ng & McLay, 2005.

Conclusions

We have herein analysed a dataset of 22,363 decapod species names spanning the period 1758–2024, to add to the knowledge base on temporal trends in species naming practices, with for the first time a largely marine taxon added. Our findings are largely in agreement with previous works, identifying an initial dominance in morphologically-derived names which transitions into an almost-equal divide across morphological, geographical, and eponymic names from the mid-20th Century. In contrast to previous studies, a quantity of eponyms honouring imperialist figures was identified only in very minor proportions. A slight gender bias was found in species named after scientists and those involved in the scientific process, once separated from the imbalance which is unavoidably reflective of the divide within the field. Though of course these artefacts should not be dismissed altogether, it is important to consider them in perspective.

Supplemental Information

Supplemental Information 1 Complete list of species-level and infrasubspecific names used in this study, with their assigned etymology classification.

The WoRMS Data Management Team is warmly thanked for database support. Cedric d’Udekem d’Acoz is acknowledged for commenting on an earlier draft.

Additional Information and Declarations

Competing Interests

The authors declare that they have no competing interests.

Author Contributions

Sammy De Grave conceived and designed the experiments, performed the experiments, analyzed the data, prepared figures and/or tables, authored or reviewed drafts of the article, and approved the final draft.

Elizabeth Cole performed the experiments, analyzed the data, prepared figures and/or tables, authored or reviewed drafts of the article, and approved the final draft.

Sancia E. T. van der Meij performed the experiments, analyzed the data, authored or reviewed drafts of the article, and approved the final draft.

Data Availability

The following information was supplied regarding data availability:

All raw data are available at DecaNet (https://www.decanet.info/index.php), a sub-portal of the World Register of Marine Species (WoRMS; www.marinespecies.org).

The species records can be retrieved using the Search Taxa option, with raw data accessible under Attributes → Etymology.

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
