# Peer review of "Decoding the bare necessities of decapod crustacean nomenclature through the ages"

_PeerJ, doi:10.7717/peerj.20337_

## Round 0.1 · original submission · Minor Revisions

· Academic Editor

Minor Revisions

The reviewers only suggested minor changes to the paper, and were overall satisfied with the quality of the manuscript.

·

Basic reporting

This article is clearly written and the literature, figures, tables, are all clear.

Experimental design

The purpose of the paper is well delineated. statistical methods were used, and interpretations of data were cross-checked among authors.

Validity of the findings

Findings are clearly reported, with many fun and interesting examples. I enjoyed reading it.
It made me think about my own naming practices, which generally fall into morphology-based (using Latin or Greek stems that are not commonly used), names of collectors or donors, and places.

Additional comments

I wonder if the data included includes fossil taxa. DecaNet has both fossil and extant taxa. Were the fossil ones used?
Some examples: Mesoparapylocheles michaeljacksoni Fraiije et al., 2012, which indeed got media attention. It was named after the singer because on the trip during which the specimen was collected, we were all eating dinner in a small restaurant. The TV was playing Michael Jackson videos, for reasons we did not understand, until finally it was reported that he had died.

Sodakus tatankayotankaensis Bishop, 1978, is a pretty long species name 919 characters), after Sitting Bull, as the specimen was from South Dakota (Sodakus) and from a reservation (I think Standing Rock).

Anyway, the paper made me remember some names that I think are interesting.
And I didn't know this was a thing we are doing now, analyzing name derivations. It's interesting.
Carrie E. Schweitzer

·

Basic reporting

Some publications referred to in the text are not listed in the References and vice versa (see file attached).

Experimental design

no comment.

Validity of the findings

No comment.

Additional comments

I have made some minor corrections in the file attached.

·

Basic reporting

I like this paper that discusses various contributing factors to the naming of species and highlights trends such as the decreasing gender bias. It is a fun paper that I think will be of interest to anyone who deals with scientific names on a regular basis. I think perhaps it could strengthen the conclusion by suggesting that we somehow capture the rationale for naming species in the naming process. I have few critical comments, just a few questions that the authors may choose to consider or ignore and highlights few minor typos.

What happens when species names change? Has the use of vulgaris declined as we have named more species, e.g. Homarus vulgaris > Homarus gammarus?
How much is the geography of names impacted by the concentration of effort of scientists/taxonomists?
Were the species discoverers (if they were not taxonomists) ever named - e.g. the victorian adventurers who collected species for money?
Have any species been names to insult someone?
Do naming conventions vary by taxonomist?
You point out that the reasons for naming species particular names are often lost to history. Would it be worth suggesting that the reasons for using particular names becomes part of the process?

86: How was Guedes headline chasing? Maybe you need "who suggested that . . . " to clarify this for non-specialists?
90: . . . conservation, cultural and dietary importance?
190: that instead of which? (possibly pedantry on my part)
199 22,363 and 11,981
203 ...referred to the morphology . .
257 . . .unattainable t. ?
287 Should these titles be in italics?
305 "Standing on the shoulders of giants"? Wasnt this first used sarcastically?

Table 1: Expedidoos??

Experimental design

No comment

Validity of the findings

No comment

Additional comments

No comment

---

## Round 0.2 · Minor Revisions

· Academic Editor

Minor Revisions

The manuscript has been reviewed again, and while it is progressing well toward acceptance, our Section Editor, Dr. Kenneth De Baets, has provided a few additional comments that we would like you to address before final acceptance. These are considered minor revisions, and we invite you to revise the manuscript accordingly. Please see Dr. De Baets’ comments below:

"The paper is really interesting, but I am surprised they focus on one headline chasing study, while not mentioning others having more nuanced views (e.g., Jiménez-Mejías et al. 2023: https://doi.org/10.1093/biosci/biae043). More importantly, as they analyse and discuss the relative contribution of female and male names over time, I am a bit surprised they do not discuss there are more than 2 sexes in nature and as aware ending of words too. Ironically, crustaceans are one of the groups with very fluent sex definitions and changes through ontogeny. I guess there may be no known examples of names relating to other sexes. Could such cases be identified using the endings o species names terms It may be difficult to assign sexes otherwise based on names as usually sex is not designated in the manuscript, and such information is not necessarily public. I am also aware that some colleagues change ending to other sex when naming sounds odd (e.g., -anus). It would be great to point that out more explicitly how to deal with such cases."

---

## Round 0.3 · accepted · Accept

· Academic Editor

Accept

Following the editor’s comments, the authors have provided a satisfactory response that did not necessitate any changes to the manuscript. As such, the paper is now accepted for publication in PeerJ.